# Ovarian Cancer and Glutamine Metabolism

**DOI:** 10.3390/ijms24055041

**Published:** 2023-03-06

**Authors:** Zacharias Fasoulakis, Antonios Koutras, Thomas Ntounis, Ioannis Prokopakis, Paraskevas Perros, Athanasios Chionis, Ioakeim Sapantzoglou, Alexandros Katrachouras, Kyriakos Konis, Athina A. Samara, Asimina Valsamaki, Vasileios-Chrysovalantis Palios, Panagiotis Symeonidis, Konstantinos Nikolettos, Athanasios Pagkalos, Sotirios Sotiriou, Marianna Theodora, Panos Antsaklis, Georgios Daskalakis, Emmanuel N. Kontomanolis

**Affiliations:** 11st Department of Obstetrics and Gynecology, General Hospital of Athens ‘ALEXANDRA’, National and Kapodistrian University of Athens, Lourou and Vasilissis Sofias Ave, 11528 Athens, Greece; 2Department of Obstetrics and Gynecology, Laiko General Hospital of Athens, Agiou Thoma 17, 11527 Athens, Greece; 3Department of Obstetrics and Gynecology, University General Hospital of Ioannina, University of Ioannina, Stavrou Niarchou Str., 45500 Ioannina, Greece; 4Department of Obstetrics and Gynecology, General Hospital of Arta, Lofos Peranthis, 47100 Arta, Greece; 5Department of Embryology, University Hospital of Larissa, Mezourlo, 41110 Larissa, Greece; 6Department of Internal Medicine, General Hospital of Larisa, Tsakalof 1, 41221 Larisa, Greece; 7Department of Obstetrics and Gynecology, University General Hospital of Larissa, Mezourlo, 41110 Larissa, Greece; 8Department of Obstetrics and Gynecology, University General Hospital of Alexandroupolis, 6th klm Alexandroupolis-Makris, Dragana Alexandroupolis, 68100 Alexandroupolis, Greece; 9Department of Obstetrics and Gynecology, Democritus University of Thrace, 6th km Alexandroupolis-Makris, 68100 Alexandroupolis, Greece; 10Department of Obstetrics and Gynecology, General Hospital of Xanthi, Neapoli, 67100 Xanthi, Greece; 11Department of Embryology, Faculty of Medicine, School of Health Sciences, University of Thessaly, 41110 Larissa, Greece

**Keywords:** ovarian cancer, glutamine, cancer cell, metabolism, cancer treatment

## Abstract

Cancer cells are known to have a distinct metabolic profile and to exhibit significant changes in a variety of metabolic mechanisms compared to normal cells, particularly glycolysis and glutaminolysis, in order to cover their increased energy requirements. There is mounting evidence that there is a link between glutamine metabolism and the proliferation of cancer cells, demonstrating that glutamine metabolism is a vital mechanism for all cellular processes, including the development of cancer. Detailed knowledge regarding its degree of engagement in numerous biological processes across distinct cancer types is still lacking, despite the fact that such knowledge is necessary for comprehending the differentiating characteristics of many forms of cancer. This review aims to examine data on glutamine metabolism and ovarian cancer and identify possible therapeutic targets for ovarian cancer treatment.

## 1. Introduction

Ovarian cancer (OC) is the seventh most prevalent cancer in women and the eighth leading cause of cancer mortality worldwide [1]. In 2014, there were 14,270 OC-related deaths in the USA, with an anticipated number of 21,980 new cases this year. Generally, 140,000 women die globally per year due to OC. Of all gynecologic cancers, OC has the highest mortality rate. The 5-year survival rate is lower than 50% with a median age of 63 years at diagnosis, and more than 70% of patients present with advanced disease. This is partly due to the fact that OC patients typically receive a diagnosis of the disease in an advanced stage, as there are no specific symptoms of OC in primary stages. In addition, there are no suitable screening procedures for OC despite recent advances in technology for more accurate diagnostic tests [2,3]. Epithelial cells are the site of genesis for the vast majority of primary ovarian malignancies; nevertheless, other ovarian non-epithelial cell types (stroma and germinal cells) are responsible for the development of 10% of all ovarian carcinomas [4]. Women with advanced epithelial OC receive chemotherapy using a platinum-based regimen combined with paclitaxel. This treatment comes after an initial effort in which the most effective cytoreductive surgery possible is performed. Despite the fact that this chemotherapy treatment is effective against the vast majority of OCs, most patients still experience drug resistance and recurrence [5,6]. Therefore, the development of cutting-edge cytotoxic treatment methods is a pressing need for the purpose of improving patient outcomes in OC [7].

Cancer cells have higher energy and biosynthetic demands for fast growth and replication. These traits help cancer cells grow. The Warburg effect is cancer cell metabolism’s most famous component. In the presence of oxygen, cancer cells create ATP and lactate by glycolyzing glucose [8,9]. Despite having high glycolysis rates, cancer cells exhibit normal mitochondrial oxidative phosphorylation (OXPHOS) and glutamine metabolism [10]. Glucose and glutamine are crucial to cancer cell proliferation because they supply bioenergetics and intermediates for macromolecular synthesis. When glucose is scarce, cancer cells convert glutamine to lactate quicker than normal cells. This gives cancer cells a glucose-free metabolic route [11]. The level of glutamine in the medium correlates with glutaminase (GS), the first enzyme in glutaminolysis [12]. Genetic and epigenetic history likely affects whether glutamine can promote cell growth [13,14]. Acivicin is an analog of glutamine that inhibits glutaminolysis in order to treat cancer in animal models [15,16,17]. Recent studies demonstrated that limiting glutamine absorption is a viable novel AML treatment [18]. However, the mechanism behind glutamine’s lack of effect on OC cell growth is unknown [19].

## 2. Materials and Methods

For this review, authors searched MEDLINE (National Library of Medicine, Bethesda, Maryland, MD, USA; January 1980 to December 2022) and the Cochrane Register of Controlled Trials (The Cochrane Collaboration, Oxford, UK). An electronic search approach included the phrases ‘ovarian cancer; glutamine; metabolism;’. To find further research of interest, references of the selected publications and review articles were evaluated. To select possibly relevant papers for this study, authors evaluated all the citations returned from the computerized search. (Appendix A, Appendix A)

## 3. Pathways of Glutamine Metabolism

The determination of the underlying molecular pathways responsible for the impact of glutamine on cell growth in OC cells and the assessment of those effects are of utmost importance. In OC cells, glutamine deprivation reduces cell growth, significantly enhances apoptosis, halts the cell cycle in the G1 phase, and increases the production of reactive oxygen species (ROS). Additionally, under conditions of cysteine deficiency, glutaminolysis promotes the fast depletion of GSH by GPX4, resulting in potent ferroptosis [20]. When glutaminolysis is blocked, the GSH turnover rate is decreased, and cysteine deficiency does not cause ferroptosis. The mTOR/S6 pathway is regulated by glutamine, which results in an increase in glycolytic activity in cells and stimulates cell proliferation. According to the results of this research, one potential technique for treating OC is to focus on cellular glutamine metabolism [21].

Furthermore, cisplatin (CDDP) significantly increases the life expectancy of cancer patients. However, a significant number of patients seem to develop cisplatin resistance, resulting in treatment failure. Understanding and thereby preventing OC cell resistance to CDDP remains a challenge. In a recent study, capillary electrophoresis-time-of-flight mass spectrometry was employed to study the metabolic profiles of cancer cells sensitive to CDDP. The metabolic profile of CDDP-resistant cells, after being compared to non-CDDP-resistant cells, was found to be different. The activation of glutamine metabolism pathways was further investigated using Western blot and real-time PCR analyses. In order to examine cellular health, a test known as the 3-(4,5-dimethylthiazol-2-yl)-2,5-diphenyltetrazolium bromide (MTT) assay was performed to measure the amount of damage caused by a toxic substance.

Glutamine is depleted in various types of cancer, especially in cancers that are poorly vascularized [19]. GS is the only enzyme responsible for creating glutamine on its own. This is important in cancer metabolism because of its assistance in tumors’ growth. GS has pro-tumoral characteristics involving the synthesis of glutamine, which, moreover, supports nucleotide synthesis. GS is expressed at a large degree in the tumor microenvironment (TME) and provides glutamine to cancer cells. Thus, it permits cancer cells to keep glutamine at a sufficient level for its catabolism [21].

Compared to A2780 cells, A2780cis cells have much higher levels of glutamine, glutamate, and glutathione (GSH), which is a key mediator of cancer cell survival and resistance to chemotherapy made from glutamate [22,23]. Moreover, glutamine deficiency results in lower glutathione (GSH) concentrations as well as CDDP resistance in A2780cis cells. In resistant A2780cis cells, glutamine synthetase (GS/GLUL), an enzyme that converts glutamate into glutamine and impairs GSH synthesis, is almost totally downregulated. These cells are resistant to the effects of the drug. Moreover, the DNA demethylating agent 5-aza-2’-deoxycytidine is used to treat A2780cis cells, which results in the restoration of GS expression and a decrease in CDDP resistance. In contrast, knocking down GS leads to an increase in CDDP resistance in A2780 cells that are CDDP-sensitive. As a result, the metabolism of glutamine may provide an opportunity as a novel therapeutic target for CDDP resistance [24].

Moreover, the possibility that glutamine significantly contributes to cell proliferation in a variety of cancer forms is of crucial importance. Through a process known as the tricarboxylic acid (TCA) cycle, their metabolism is reprogrammed to use glutamine. Cancer cells are dependent on glutamine for survival. Isotope tracers and bioenergetic analysis are used to determine whether low-invasive OC cells need glutamine to the same degree that high-invasive OC cells do. The microarray data from OC patients reveals that those with glutaminolysis have a lower possibility of survival. There seems to be a connection between gene expression and glutamine synthesis, especially regarding glutamine breakdown being a possible predictor of a patient’s response to glutamine supplements. This is essential, as it implies sustainability. It was also found that glutamine regulates the activation of the signaling pathway mediator STAT3, which is eventually responsible for the modulation of cancer hallmarks in invasive OC cells. As a result, it is possible that invasive OC therapy could be based on glutamine metabolism. It was suggested that a potential therapeutic strategy for treating OC may involve focusing on high-invasive OC cells by preventing glutamine from entering the TCA cycle (Appendix A). Another strategy may involve focusing on low-invasive OC cells by reducing glutamine output and STAT3 expression [19,24].

Additionally, the effects of combined chemotherapy with the glutaminase inhibitor bis-2-(5-phenylacetamido-1,3,4-thiadiazol-2-yl) ethyl sulfide (BPTES) on OC cells that are resistant to the effects of chemotherapy was investigated. By reducing cell growth, treatment with BPTES makes chemotherapy-resistant cancer cell lines susceptible to paclitaxel and cisplatin. This effect occurs regardless of the cell lines’ dependence on glutamine. BPTES monotherapy significantly decreases the capacity of glutamine-dependent cancer cells in order to establish colonies in a clonogenic assay. Glutaminase 1 (GS1) isoforms KGA and GAC are much more abundant in metastatic glutamine-dependent cancer cells than in untransformed cells. Additionally, the use of siRNA to target both isoforms together makes cancer cells more susceptible to cisplatin than the use of either GAC or KGA alone. The aforementioned findings indicate that both GS1 isoforms must be targeted in the treatment of metastatic OC because both are critical for survival in glutamine-dependent OC [25].

A comparative data analysis of gene expression between cancer cells and normal control tissues from 11 distinct types of cancer was conducted in order to comprehend glutamine and glutamate metabolic cancer rates. More specifically, a linear regression model was used to investigate the contribution of glutamine and glutamate to each of the seven biological processes in cancer tissues compared to normal tissues. Although a number of the computational forecasts were consistent with previously collected information, they arrived at the original hypothesis that the contribution of glutamine to nucleotide synthesis in cancer is negligible, if not nonexistent; in contrast to bladder and lung tumors, glutamine is not involved in the formation of asparagine in cancer, and glutamate is not implicated in the synthesis of serine. This hypothesis might result in the creation of innovative treatment approaches focusing on glutamate and glutamine metabolism [26].

Cancer cells seem to rely heavily on glutamine for the production of their harmful products. Cell cycle progression, apoptosis, cytotoxicity, cellular stress, and glucose/glutamine metabolism were examined in order to ascertain whether the OC cell lines HEY, SKOV3, and IGROV-1 rely on glutamine. Each of the three different OC cell lines responded differently to glutamine therapy, with each line experiencing an increase in cell proliferation proportional to the amount of glutamine they received. Endoplasmic reticulum stress proteins and reactive oxygen species levels are proven to rise as a consequence of glutamine availability reduction. Thus, this leads to an increase in the number of cells that are unable to survive or grow. Glutamine supplementation enhances the activity of glutaminase (GS) and glutamate dehydrogenase, both of which are involved in glutamine production, due to the modulation of mTOR/S6 and MAPK pathways. A drop in glutamate and glutamine levels downregulates the activity of mTOR, which, in turn, inhibits cell proliferation through the inhibition of S6 expression or rapamycin treatment. As a result, it might be viable to target glutamine metabolism as a therapeutic strategy for OC therapy [7].

It is supported that glutamine’s participation in cancer is influenced by a number of factors. These factors include the type of tissue, the underlying cancer genetics, the microenvironment of the tumor, and other variables such as food and host physiology. Thus, glutamine requirements in cancer are relatively variable overall [27]. The growth of cancer cells is controlled by reactive stromal cells, which are an essential part of the tumor microenvironment (TME). The stromal targets that make cancer cells vulnerable have been elusive up to this time, despite the fact that targeting stromal cells might be a potential therapeutic path in controlling communication between the tumor microenvironment (TME) and cancer cells. A new mechanism for reprogramming the metabolic pathways of reactive stromal cells via an enhanced glutamine anabolic pathway was recently discovered. Defective stromal metabolism grants these cells unique metabolic flexibility and adaptation mechanisms. Thus, stromal cells use carbon and nitrogen from noncoding sources to synthesize glutamine in the midst of dietary deprivation. Using an orthotopic mouse model for ovarian carcinoma, a reduction in tumor mass, nodules, and metastases was observed by concurrently inhibiting glutamine synthetase in the stroma and glutaminase in cancer cells. The purpose was the production of beneficial therapy outcomes [28].

Notably, the understanding of OC has significantly advanced because of recent breakthroughs in molecular genetics on redox homeostasis. Cancer is often distinguished by increased levels of reactive oxygen species (ROS) as well as the activation of antioxidant genes. ROS are responsible for producing DNA damage and mutations, genomic instability, and aberrant anti-tumorigenic and pro-tumorigenic signaling, all of which play a significant role in the progression and suppression of tumors. Cancer cells are able to produce more antioxidants as a defense mechanism against ROS that are produced in excess. Redox equilibrium is also promoted in OC by antioxidants such as CD44 variant isoform 9 (CD44v9) and nuclear factor erythroid 2 related factor 2 (Nrf2). In addition, research conducted on a wide variety of cancers has shown that antioxidants have a dual role in the formation of tumors as well as in their suppression. On the other hand, the genetic loss of antioxidant activity is unable to prevent the formation and progression of cancer in animal models. In order to stop carcinogenesis, host-derived antioxidant systems are required; this demonstrates that antioxidants are an essential component in the fight against the development of cancer. Moreover, the activation of antioxidants causes cancer cells to take on more aggressive traits. Antioxidant inhibitors may hasten the death of cancer cells by increasing intracellular levels of reactive oxygen species (ROS). Apoptosis, elevated chemosensitivity, and increased anticancer activity may be induced by inhibiting CD44v9 and Nrf2 at the same time via a route triggered by reactive oxygen species (ROS). Antioxidants were shown to have effects that are both pro- and antitumor. The development of antioxidant-specific inhibitors and a clearer comprehension of the impact that antioxidants play in preserving redox homeostasis are both essential stages in OC therapy [29].

Furthermore, the metabolic processes of cancer cells may be described as an intricate and ever-changing network of regulated pathways. This network could use high-throughput screening methods more beneficially if they were more rapid, sensitive, and adaptable to high-throughput formats. Thus, tests must involve little sample preparation and be scalable to well sizes of 384 while also requiring less throughput and automation [30].

Bioluminescent assays for the detection of glutamine, glutamate, lactate, and glucose that are suitable for high-throughput research on glycolysis and glutaminolysis were discovered. These are two of the most essential metabolic processes that occur in cancer cells. A sensitivity of 1–5 pmol/sample, a broad linear range of 0.1–100 M, and a huge dynamic range of more than 100-fold for the purpose of evaluating both intracellular and extracellular metabolites are required. Importantly, the tests include the rapid shutdown of endogenous enzymes, which eliminates the need for deproteinization phases that were necessary in earlier methods. As a model system, OC cell lines were used for the tests, which allowed monitoring glucose and glutamine intake as well as lactate and glutamate release over time. Assays for lactate and glutamate that use homogenous formats are reliable (Z’ = 0.6–0.9) and capable of being multiplexed with real-time viability tests to provide evidence that they are internally controlled. These experiments are utilized to screen a small compound library, and the results reveal compounds that inhibit and promote the generation of glutamate and lactate, respectively [30].

One of the most distinctive features of OC is a rise in cellular metabolism, which is mainly driven by glutamine and glucose. Targeting metabolic pathways is an intriguing and potentially fruitful technique for improving therapy effectiveness and reducing drug resistance. In response to platinum treatment, glucose and glutamine metabolisms were initially enhanced in platinum-sensitive OC cell lines. In platinum-resistant cells, glutamine ASCT2 and glutaminase transporter expression was increased, showing a substantial dependency on the availability of glutamine. In comparison to platinum-sensitive cell lines, this led to a higher oxygen consumption rate, which indicated a stronger dependency on glutamine consumption through the tricarboxylic acid cycle. The relevance of glutamine metabolism is shown by the fact that platinum resistance may be achieved by maintaining high levels of glutaminase expression. Re-sensitization of platinum-resistant cells to the effects of platinum treatment was achieved by inhibiting glutaminase activity using shRNA. Importantly, when examined in vitro, the combination of the glutaminase inhibitor BPTES with platinum resulted in a synergistic decrease in the growth of both platinum-sensitive and platinum-resistant ovarian tumors. When compared to individual therapies, the combination of platinum and BPTES led to a significant increase in the amount of apoptosis that was induced. Targeting glutamine metabolism in conjunction with chemotherapy based on platinum appears to be a potential therapeutic strategy, especially for the management of drug-resistant OC [31].

Myeloid-derived suppressor cells are a diverse group of immune-regulating cells. They are also referred to as immature CD11b(+)Gr1(+) myeloid cells, which develop immunosuppressive properties. A study aimed to determine how the surrounding environment of OC affects the immunosuppressive impact of CD11b(+)Gr1(+) myeloid cells. The mouse model of intraperitoneal ID8 syngeneic epithelial OC served as the basis for each experiment that was carried out. Immunotherapy and myeloid cell suppression were both delivered using various treatments such as anti-Gr1 mAb, gemcitabine therapies, and/or anti-PD1 mAb. The efficacy of the therapy was assessed by utilizing histology, in situ luciferase-guided imaging, and survival curves. Adoptive transfer experiments were carried out using mice that were congenic for either CD45.2 or CD45.1. The evaluation of immunological surface and intracellular markers was accomplished using flow cytometry. The levels of protein and RNA expression of various indicators were analyzed by using techniques such as RT-PCR, Western blot, and ELISA. Ex-vivo investigations used myeloid cells derived from bone marrow as a source of cells to work with [32].

The data demonstrate that the elimination of Gr1(+) immunosuppressive myeloid cells, alone or in conjunction with anti-PD1 treatment, is a potent inhibitor of OC growth. These results show that immunosuppressive CD11b+Gr1+ myeloid cells might play a part in the process of developing OC. According to the findings of a mechanistic study, ID8 tumor cells and the environment in which they were found created factors that were both recruiting and regulating for immunosuppressive CD11b+Gr1+ myeloid cells. Myeloid cells that were stimulated by ID8 tumors indicated elevated immunosuppressive marker expression and adopted a metabolic profile that was characterized by enhanced oxidative phosphorylation driven mostly by glutamine. This metabolic profile was characteristic of myeloid cells that had been stimulated by ID8 tumors. When the glutamine metabolic route was blocked, oxidative phosphorylation was also halted, and the generation of immunosuppressive markers and their activity were reduced. It was shown that tumor-primed myeloid cells had the highest expression levels of the KGDC subunit dihydrolipoamide succinyl transferase (DLST), which is involved in the TCA cycle. This gene was identified as the gene with the highest expression level. The inhibition of DLST caused a decrease in the oxidative phosphorylation of myeloid cells as well as the production and function of immunosuppressive markers [32].

This study revealed that the immunological environment of OC may be altered by the immune microenvironment, which, in turn, can impact the metabolism and activity of immunosuppressive CD11b+Gr1+ myeloid cells. Because the glutamine metabolism of immunosuppressive myeloid cells was targeted with DLST, the activity of these cells was decreased, which led to a tumor microenvironment that was less immunosuppressive. Therefore, the inhibition of glutamine metabolism might potentially boost the effectiveness of immunotherapy in treating OC [32].

Amide Valley’s effect on paclitaxel-treated OC mice’s immune function was recently studied. Fifty female SPF BALB/c mice were used as test subjects and split into four categories: normal control, tumor control, paclitaxel, and glutamine. Forty mice were given OC, with the normal control group being the exception. The control group received 25 mg/kg of saline, whereas the paclitaxel group received the same number of intraperitoneal injections of paclitaxel, 25 mg/kg glutamine, and 25 mg/kg glutamine + paclitaxel. Control mice were not treated. After 2 weeks of therapy, the CD3+, CD4, CD4+, CD8, and CD8+ T cell subsets of each mouse group were identified by using flow cytometry, and their CD4/CD8 ratios were determined. OC prevalence in mice was 95% (38/40). The normal control group’s hair, food, excretion, and activity were normal, whereas the other three groups of OC mice had thin, dull hair, a decreased diet, and slower activity. When contrasted with the mice in the normal control group, the body mass of the other three categories of OC mice rose dramatically (*P* = 0.05) prior to treatment due to the tumor body. However, following treatment, the body mass of the mice in the combined intervention group declined substantially (*P* = 0.05), indicating the success of the combined intervention. CD3+, CD4, CD4/CD8, and CD8+ levels in the tumor control and paclitaxel groups differed significantly (*P* = 0.05). The tumor control group and paclitaxel group had similar CD3+, CD4/CD8, and CD8+ levels. Paclitaxel does not boost OC patients’ immunity [33,34].

Glutamine pathways have proven to be insufficiently studied in OC cells. This amino acid is widely distributed in tissue and plasma while being a fundamental energy source for cancer cells at the same time. Reducing the production of cancer cells is the goal of glutamine-induced cell proliferation in female OC cells. Molecular intervention, which is meant to metabolize glutamine, might prove effective in the treatment of OC. Glutamine also seems to protect cells against stress [35,36].

Glutamine deficiency can increase the amount of glutamate and ketoglutarate in the TCA cycle, which can lead to increased aerobic glycolysis. Glutamine degradation reduces glucose and ATP consumption in three OC cell lines, but it increases glutamine expression and glycolitextane protein 24 h later. Glutamine levels in the blood affect each cell line differently. Cancer cells use glucose and glutamine to fuel their growth. Glutamine may help protect cells from glutamine deficiency or other stressors. In order to maintain adequate levels of glutathione, ketoglutarate, glycogen, and ROS in cells, it is important to reduce Gls2 expression [35].

Glutamine’s carbon backbone is oxidized in cancer cells to provide energy for growing cells. This mechanism is a major energy source for cancer cells. Glucose ability to lead to the resolution of phosphate compounds is related to the ability of mitochondria to process oxaloacetate esters. As cancer cells undergo changes in their microenvironment and genetic makeup, their glycolysis and oxidative phosphorylation balance may shift. This may lead to a modification in the cancer cells’ energy production, affecting their survival and growth. Cancer’s energy sources drive cells to use phosphorus efficiently, resulting in cell damage and cancer. Cells can rapidly change between dialysis and oxidation as they encounter different sources of cancer energy [36].

Glutamine deprivation leads to the reduction of available glutamate and of the TCA cycle, which increases aerobic glycolysis. In one study, glutamine degradation decreased glucose and ATP uptake in three lines of OC cells, although enhancing glutamine expression time-dependently, and glutamine-generated glycolitextane protein varied 24 h after treatment. Cancer cells may coordinate glucose and glutamine metabolism due to genetic alterations [37]. Mutations that increase proliferative signaling induce OC carcinogenesis. These pathways appear significant in glutamine-induced regulation of OC proliferation, invasion, and bioenergetics. Intestinal permeability, protein synthesis, and pTr cell proliferation are all controlled by the mTOR/S6 signaling system [38,39,40].

There is a mutation in oncogenic RAS present in 15% of cases of ovarian cancer [41]. At the same time, it was demonstrated that cancer with a mutant KRAS upregulates the activity of NRF2, which leads to the stimulation of glutamine metabolism [42]; this ultimately results in the formation of chemoresistant tumors. As a result, the potential value of using glutaminase inhibitors in conjunction with chemotherapies as a combined treatment strategy is highlighted. Moreover, in ovarian cancer, oncogenic RAS is responsible for maintaining the redox equilibrium within the cell, though it is essential to point out that NRF2 plays a unique part in the development of ovarian cancer [43,44].

Concerning the clinical importance of glutamine metabolism in ovarian cancer, two clinical trials are included in this review. First, CB-839 is an orally administered, small-molecule GLS inhibitor based on BPTES [45]. CB-839 has a modest inhibitory effect on the majority of malignancies in a phase I clinical trial. It is severely decreased, however, when used with other anticancer medications. Numerous Phase I/II clinical trials have demonstrated that CB-839 is well-tolerated by patients. It can enhance the efficacy of capecitabine, cabozantinib, and other medications in suppressing glutamine-dependent cancers. The treatment of mice with ovarian cancer with CB-839 extends their lifespan [46]. In an Arid1a-deactivated mouse ovarian cancer xenograft model, CB-839 in combination with the immune checkpoint drug anti-PDL1 demonstrated synergistic benefits [47]. A CB-839 and niraparib clinical trial for platinum-resistant ovarian cancer patients (NCT03944902) was identified. Due to the company’s decision to discontinue use of this medicine, this trial was terminated and will not be restarted. One person dropped out of the study. Another phase I trial examines the safety, adverse effects, and optimal dose of the agent IACS-6274 with or without bevacizumab and paclitaxel in the treatment of patients with metastatic solid tumors. IACS-6274 may inhibit the growth of the tumor cells by inhibiting essential enzymes for cell proliferation. IACS-6274 administered with or without bevacizumab and paclitaxel may aid in disease management (Appendix A).

## 4. Conclusions

Overall, it was found that glutamine has an essential impact on the OC cell cycle, apoptosis, proliferation, glycolysis, cell stress, and energy flow as well as on the mTOR and S6 signaling pathways. Due to its ability to stimulate the mTOR/S6 and MAPK pathways, glutamine may aid in promoting cell proliferation. Focusing on glutamine metabolism may be a helpful new approach for OC.

Furthermore, it was discovered that glutamine metabolism is crucial for the emergence of CDDP resistance in OC cells. A2780cis cells had substantially larger levels of glutamate, glutamine, and glutathione (GSH) than A2780 cells do. In A2780cis cells, the expression of GS was drastically reduced. In addition, the introduction of 5-aza-dC into A2780cis cells resulted in the restoration of GS expression and a reduction in resistance to CDDP. In many different types of cancer, using DNA methyltransferase inhibitors to target glutamine metabolism may be an effective way to overcome chemotherapy resistance. 

## Data Availability

No new data were created or analyzed in this study. Data sharing is not applicable to this article.

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
