# Peer review of "Ovarian Cancer and Glutamine Metabolism"

_ijms, 2023, doi:10.3390/ijms24055041_

Round 1
Reviewer 1 Report (New Reviewer)
Review article by Dr. Kontomanolis and group elaborating on the role of glutamine metabolism in ovarian cancer and exploring the therapeutic opportunities for suffering patients. This a novel topic for review article, which is very informative and planned down thoughtfully. Though few things need to be addressed before it is ready for acceptance, they are as follows:
1. Authors must change the format or style of the writing. For every other study, they are not required to mention "according to .....et al." They can say the work but use only reference section for the referring to anyone's work. Otherwise, it doesn't look delightful and not worthwhile for reading purposes also.
2. In ovarian cancer, oncogenic RAS is mutated with 15% frequency (PMID: 33870211). At the same time, it has been shown that mutant KRAS cancer upregulates NRF2's activity, resulting in induction of glutamine metabolism (PMID: 31911550) which creates chemoresistance. This suggests the role of utilizing glutaminase inhibitors as a combination therapy with chemotherapies. In ovarian cancer also oncogenic RAS preserves intracellular redox balance(PMID: 31000598). Additionally, it must also be noted that NRF2 plays a distinct role in ovarian cancer (PMID: 35453348). Authors must include few lines discussing the above-mentioned topics.
3. Authors should add a list of clinical trials in ovarian cancers targeting cancer metabolism. This should be helpful for readers to understand the clinical impact of this manuscript.
4. Authors should modify the figure 2 by adding the enzyme names and also changing the color of boxes or making it more professional representation. Authors must work on this. Check any IJMS review article to get an idea about it.
Author Response
Dear Reviewer,
Thank you very much for your kind remarks.
- We changed the format and style of the writing. We have deleted "according to .....et al.", and used only reference section.
- We included a few lines discussing the topics you have suggested and added the references. (PMID: 33870211, 31911550, 31000598, 35453348)
- We added a list of clinical trials in ovarian cancers targeting cancer metabolism.
- We modified figure 2 by adding the enzyme names and also changing the color of boxes.
Thank you very much in advance,
Yours sincerely,
Antonios Koutras, MD, MSc, PhDc
Reviewer 2 Report (New Reviewer)
The authors well reviewed on ovarian cancer and glutamine metabolism, and this manuscript is very interesting. My comments are as follows.
1)The authors described that glutamine is depleted in various types of cancer, especially in cancers that are poorly vascularized. Could you present the studies on which this is based? Increased glutathione has been reported in various cancer types. There are several studies reported that high levels of glutathione (GSH) observed in various types of cancer promote cancer-cell survival and resistance to chemotherapy (Estrela JM, Crt Rev Clin Lab Sci 2006, Trachootham D, Nat Rev Drug Discov. 2009).
2)Dose glutamine depletion enhance only apoptosis in OC cells? Dose it induce ferroptosis?
Author Response
Dear Reviewer,
Thank you very much for your kind remarks,
1) We referenced the study mentioning that glutamine is depleted in various types of cancer, especially in cancers that are poorly vascularized. Furthermore, we added the suggested studies mentioning that high levels of glutathione (GSH) observed in various types of cancer promote cancer-cell survival and resistance to chemotherapy (Estrela JM, Crt Rev Clin Lab Sci 2006, Trachootham D, Nat Rev Drug Discov. 2009).
2) We have added information about glutamine depletion enhancing not only apoptosis in OC cells, but also inducing ferroptosis.
Thank you very much in advance.
Yours sincerely,
Antonios Koutras, MD, MSc, PhDc
Round 2
Reviewer 1 Report (New Reviewer)
All concerns have been addressed, ready for acceptance.
This manuscript is a resubmission of an earlier submission. The following is a list of the peer review reports and author responses from that submission.
Round 1
Reviewer 1 Report
This article is very poorly written with numerous grammatical errors, broken sentences and confusing statements. It is impossible to review this article in its current form unless the language is revised by an expert. Moreover in the last few years there have been so many review articles on glutamine metabolism in cancer. Therefore the authors need to clearly present their observations and key points they want to convey based on existing literature. Also authors should summarize the contents using a Figure or Tables based on literature related to glutamine metabolism in ovarian cancer.
Reviewer 2 Report
Manuscript by Fasoulakis et al discussed the role of glutamine metabolism in cancer cells and tumor progression. However, manuscript is not well written and some of the text has uncanny resemblance to text within already published study. I would suggest not to publish this manuscript in current format.